# Preprocessing for Keypoint-Based Sign Language Translation without Glosses

**DOI:** 10.3390/s23063231

**Published:** 2023-03-17

**Authors:** Youngmin Kim, Hyeongboo Baek

**Affiliations:** Department of Computer Science and Engineering, Incheon National University (INU), Incheon 22012, Republic of Korea

**Keywords:** computer vision, deep learning, sign language translation, video processing

## Abstract

While machine translation for spoken language has advanced significantly, research on sign language translation (SLT) for deaf individuals remains limited. Obtaining annotations, such as gloss, can be expensive and time-consuming. To address these challenges, we propose a new sign language video-processing method for SLT without gloss annotations. Our approach leverages the signer’s skeleton points to identify their movements and help build a robust model resilient to background noise. We also introduce a keypoint normalization process that preserves the signer’s movements while accounting for variations in body length. Furthermore, we propose a stochastic frame selection technique to prioritize frames to minimize video information loss. Based on the attention-based model, our approach demonstrates effectiveness through quantitative experiments on various metrics using German and Korean sign language datasets without glosses.

## 1. Introduction

Effective communication is essential for full participation in society, yet deaf individuals often face challenges in accessing information and communicating with hearing unimpaired individuals [1]. Sign language is a visual language that provides the primary means of communication for deaf individuals. It uses a combination of hand movements, facial expressions, mouth movements, and upper body movements to convey information. However, sign language is often not well understood by those unfamiliar with it, making it difficult for deaf individuals to communicate with the general population. In order to address this issue, research on sign language translation (SLT) using computers has been conducted for some time to facilitate communication between deaf and non-deaf people via translating sign language videos into spoken language [2,3].

Most existing deep-learning-based SLT methods rely on gloss annotations, a minimum vocabulary set of expressions that follow the sign language sequence and grammar, to translate the sign language into spoken language. However, collecting gloss annotations can be costly and time-consuming, and there is a need for SLT methods that do not require them. The state-of-the-art study [4] attempted to perform translation without glosses, but came across two major limitations. Firstly, the translation was based on the features of the image, which resulted in it being influenced by the background rather than focusing solely on the signer. Secondly, dimensionality reduction was applied to the features through embedding to fix the input size of the model. However, this may result in the loss of important information in sign language videos.

To address the first issue, we propose a sign language translation method based on the body movement of the signer. Sign language represents a visual language essential for communication among the deaf population, and it uses various complementary channels to convey information [5]. These channels include hand movements and nonverbal elements such as the signer’s facial expressions, mouth, and upper body movements, which are crucial for effective communication [6]. Therefore, the meaning of sign language can vary depending on the position and space of the signer’s hand. Taking these factors into consideration, we focus on the signer’s movement by accurately capturing their body’s movement using keypoints, namely, specific points on the body that are used to define the movement of the signer. These keypoints are defined as the joint points of these signers.

To process the frames for SLT based on the signer’s skeleton, we must address the issue of keypoint vectors that vary significantly depending on the angle and position of the signer in the frame. To address this issue, we propose a simple but effective method. First, we propose a normalization method that uses the distance between keypoints and applies different normalization methods according to the part of the body, which we call “Customized Normalization”. This normalization method is robust to variations in the signer’s body and helps improve the performance of the translation system.

To solve the second issue of previous studies [4], we propose selecting and augmenting frames based on probabilities. Our method can reduce the loss of image information due to embedding by prioritizing frames in sign language images. Previous keypoint-based studies [7,8,9,10] have also employed various methods to fix the length of the input keypoint vector. However, these studies either use frame sampling to lose information or use only augmentation, leading to increased memory usage and slower processing time. Therefore, we propose the“Stochastic Augmentation and Skip Sampling (SASS)” method that adjusts to the length of a dynamic video frame by simultaneously using both augmentation and sampling techniques based on the length of the video.

In this paper, we apply the Sign2Text method to the dataset without gloss and propose a frame-processing method that can perform well even in cases where the video resolution differs. For sign language translation, we use the Gated Recurrent Unit (GRU) [11]-based Sequence-to-Sequence (Seq2Seq) model [12] with the Attention of Bahdanau et al. [13]. An overview of our framework is illustrated in Figure 1, and the contributions of this paper are as follows.

•We propose a new normalization method most suitable for sign language translation in keypoint processing.•We propose a stochastic frame selection method that prioritizes frames for videos of different lengths to consider the characteristics of a sign language video.•Our method is versatile because it can be applied to datasets without gloss.

The remainder of this paper is structured as follows: Section 2 reviews existing research on SLT and video processing methods. Section 3 presents our proposed video processing method and SLT approach. Section 4 presents the results of experiments evaluating the effectiveness of our method. Section 5 presents the results of the ablation study. Finally, Section 6 provides concluding remarks and suggestions for future work.

## 2. Related Works

### 2.1. Sign Language Translation

As mentioned earlier, research on sign language translation has been steadily progressing, but there are several reasons for the slow development of sign language translation systems. First, there needs to be more data for sign language translation. Since the sign language differs depending on the country and region, the data to translate a specific language differs. Therefore, it is necessary to collect sign language datasets directly by country, but collecting many datasets takes much work. In previous studies that only dealt with translating between sign language and text, the average number of words in the dataset was very small, about 3000, even though no image information was used [14,15,16]. Fortunately, researchers began building many sign language datasets as algorithms evolved. In particular, with the development of algorithms for weakly annotation data [17,18] and the development of algorithms for pose estimation [19,20], researchers began to feel the need for sign language datasets.

As a result of these developments, German sign language datasets called RWTH-PHOENIX-Weather 2012 [21], RWTH-PHOENIX-Weather 2014 [22], RWTH-PHOENIX-Weather 2014-T [4], an American Sign Language (ASL) dataset ASL-PC12 [23], and Korean sign language dataset KETI [7] were also created.

The establishment of various datasets has led to a surge in deep learning-based SLT research. The PHOENIX dataset, which includes glosses, has been particularly popular and generated much interest and research. Most of these studies extract sign features from sign language videos using Convolution Neural Networks (CNNs) [24]. For example, Yin et al. [25] combined the STMC [26] module, which extracts the signer’s features and separates body parts, with a Transformer [27] based on CNN. Camgoz et al. [4,28] also used a CNN-based model to extract the signer’s features from videos.

However, the signer’s features can also be extracted from sign language videos based on their skeleton movements, not just using CNN. Gan et al. [10] translated sign language from the PHOENIX dataset using GCN [29], based on the signer’s keypoints. Additionally, research by Ko et al. [7] and Kim et al. [8] also translated Korean sign language based on the signer’s keypoints.

### 2.2. Video Processing

Video processing is mainly applied in action recognition tasks and video recognition tasks, with video processing having a significant impact on performance [30,31,32]. Gowda et al. [33] proposed a score-based frame selection method and achieved state-of-the-art results for the UCF-101 dataset [34]. Karpathy et al. [35] performed video classification on a large-scale dataset using a video-processing method that treats all videos as fixed clips. Therefore, video processing is a crucial method for performance improvement.

Video processing mainly uses an approach that changes the number of frames, via augmenting the number of frames by transforming the frame image [36,37] or augmenting the manipulation of the order of the frames [38]. Recently, some studies have proposed a frame augmentation method using Generative Adversarial Networks (GAN) [39] or genetic algorithms for video processing using deep learning [40]. In addition to augmentation, a frame sampling method, frame skipping [41], has also been proposed.

The frame-processing method was employed in both the SLT and the action recognition. Park et al. [9] augmented the data in three ways: camera angle transformation, finger length transformation, and arbitrary keypoint removal. Ko et al. [7] used a new skip sampling method to augment the number of data by adding randomness.

## 3. Proposed Method

In this section, we introduce the overall architecture of the proposed method. Our architecture is decomposed into four parts: Extraction of keypoint, keypoint normalization, Stochastic Augmentation and Skip Sampling (SASS) and sign language translation model. The overall flow of our proposed architecture is described in Figure 2.

Camgoz et al. [4] underwent the embedding process after extracting the feature of the sign language video frame through CNN. It aims to reduce the dimension by embedding the feature point of the large amount extracted from the learning image. However, all the features extracted can include backgrounds, which can affect learning by becoming noise. Therefore, in this paper, instead of extracting features through CNN, we propose a method of extracting the keypoint of the signer and using it as a feature value. Using keypoint, we can construct a robust model against the background by only looking at the movement. Initially, the location of keypoints is taken into consideration, followed by the proposal of Customized Normalization that emphasizes hand movement, an essential aspect of sign language. In many languages, verbs and nouns, the core of a sentence, are placed in the middle. Furthermore, there are cases in which hands do not appear at the beginning and end of the video. Therefore, we define the ’key frame’ as the frame in the middle part of the video. We propose a stochastic augmentation method that enables the model to better learn essential parts of sign language videos by assigning stochastic weights to ’key frames’ of videos. Additionally, we propose Stochastic Augmentation and Skip Sampling (SASS), which combines sampling and augmentation to apply to dynamic video frame length. Finally, we perform sign language translation by applying the method of Bahdanau et al. [13] of the NMT task.

### 3.1. Extraction of Keypoint

To minimize the impact of the surrounding background on the video and focus solely on the signer, we extracted the signer’s keypoint using a skeleton point extraction method. Our approach involved using Alphapose, a framework developed by Fang et al. [20] that employs a top-down detection technique for signers and then extracts keypoints from cropped images. We used a pretrained model from the Halpe dataset [20], which has 136 keypoints. By removing the keypoints of the lower body and face keypoints to construct the keypoint vector $V=(v_{1},\cdots ,v_{i},\cdots ,v_{55})$. The location of these keypoints are shown in Figure 3.

### 3.2. Customized Normalization

We propose customized normalization according to body parts considering skeleton positions. Since the body length varies depending on the person, a normalization method considering this is needed. Although Kim et al. [8] mentioned the robust keypoint normalization method, it cannot be said to be a normalization method considering all positions because the reference point was set in some parts, not all parts. Therefore, we propose a normalization method that can further emphasize the location of each skeleton and also the movement of the hand.

We aim to normalize the keypoint vector, *V*. The *V* is divided into *x* and *y* coordinates, $V_{x}$ and $V_{y}$, and normalized, respectively. First, we designated reference points for each body part according to the face, upper body, left arm, and right arm. The reference point is defined as $r=(r_{x},r_{y})$. The corresponding keypoint numbers according to each body part are shown in Table 1, and the names corresponding to each number can be referred to in Figure 3. Furthermore, the representative value for the entire body of the signer is defined as center point, $\left(c_{x},c_{y}\right)$, and follows:(1)$$\begin{array}{c} c_{x}=\frac{1}{55}\sum_{i=1}^{55}v_{i}^{x},\;c_{y}=\frac{1}{55}\sum_{i=1}^{55}v_{i}^{y}\\ where,\;v^{x}\in V_{x},\;v^{y}\in V_{y} \end{array}$$

After obtaining the Euclidean distance, $d_{point}$, between the center coordinate, *c*, and the reference point, *r*, of each part, it is normalized based on it. It follows:(2)$$\begin{array}{c} S_{point}\;=\;\left\{\;Nose,\;Neck,\;LElbow,\;RElbow\;\right\}\\ d_{point}\;=\;\sqrt{{\left(c_{x}\;-\;r_{x}\right)}^{2}\;+\;{\left(c_{y}-r_{y}\right)}^{2}}\;\;\;where,\;\left(r_{x},r_{y}\right)\;\in S_{point}\\ \left({\bar{V}}_{point}^{x},\;{\bar{V}}_{point}^{y}\right)\;=\;\left(\frac{V_{x}-c_{x}}{d_{point}},\;\frac{V_{y}-c_{y}}{d_{point}}\right) \end{array}$$

Therefore, the keypoint for the body excluding both hands follows, $\left({\bar{V}}_{point}^{x},\;{\bar{V}}_{point}^{y}\right)$.

Next, we apply min-max scaling to the keypoints of both hands in order to make them equal in scale and compare their movements equally. Min-max scaling adjusts the range to [0,1] by dividing the difference between the maximum and minimum values of the coordinates by subtracting the minimum values from the corresponding coordinates. Here, to avoid encountering the vanishing gradient problem caused by a minimum value of 0, the range was adjusted to [0.5,0.5] by subtracting −0.5. Therefore, we define the normalized hand keypoint vectors, ${\bar{V}}_{hand}^{x}$ and ${\bar{V}}_{hand}^{y}$, by min-max scaling the keypoint and subtracting 0.5 from it, which follows:(3)$$\left({\bar{V}}_{hand}^{x},{\bar{V}}_{hand}^{y}\right)=\left(\frac{V_{x}-V_{min}^{x}}{V_{max}^{x}-V_{min}^{x}}-0.5,\frac{V_{y}-V_{min}^{y}}{V_{max}^{y}-V_{min}^{y}}-0.5\right)$$
where, $V_{max}$ and $V_{min}$ are the maximum and minimum values of the hand keypoint, respectively. Consequently, each and point subjected to our normalization method follows vectors, $V_{x}^{*}=\left[{\bar{V}}_{point}^{x};{\bar{V}}_{hand}^{x}\right]$ and $V_{y}^{*}=\left[{\bar{V}}_{point}^{y};{\bar{V}}_{hand}^{y}\right]$, and the final keypoint vector is a concatenation of them, $V^{*}=\left[V_{x}^{*};V_{y}^{*}\right]$.

### 3.3. Stochastic Augmentation and Skip Sampling (SASS)

To train a deep learning model, it is necessary to fix the input size. Previous studies have used embedding to reduce the dimension and fix the size of the input. However, this can result in the loss of important information from the video. Therefore, we propose a method that preserves the information in the sign language video without using embedding. We achieve this by adjusting the input value length through augmentation or sampling the frames. We propose the SASS (Stochastic Augmentation and Skip Sampling) method, which is a general technique that can be applied to various datasets and emphasizes the key frames of sign language videos without losing them. This is achieved through a combination of an augmentation method that considers frame priority and a sampling method that reduces frame loss. We define the *j*-th training video as $x_{j}={\left\{x_{t}\right\}}_{t=0}^{T}$ and *L* as the number of training videos, where j∈[1,L]. We select *N* frames from between the frames of training video $x_{j}$, which follows:(4)$$N=\frac{1}{L}\sum_{j=1}^{L}x_{j}$$

If the number of video frames $x_{j}$ is bigger than *N*, the sampling is used, and if the number of the video frame is less than *N*, the number of frames in each video is adjusted to *N* using augmentation. Therefore, input value, *X*, follows $X\in R^{L\times N\times |V^{*}|}$. Our method can prevent the loss of video information, which is a disadvantage of frame sampling, and the disadvantage of frame augmentation, which can prevent a lot of memory consumption. The overall flow of our SASS method follows Figure 4.

#### 3.3.1. Stochastic Augmentation

Our proposed method does not follow uniform distribution when selecting the frame but rather reconstructs the key frame into a probability distribution that can be preferentially extracted with a high probability of being selected.

The probability distribution that increases the priority for the key frame is constructed from a combination of binomial distributions. The frame selection probability set, $F_{p}$, has a size equal to the length of the video, *T*, and follows:(5)$$F_{p}=\left\{f\left(1;T,p\right),f\left(2;T,p\right),\cdots ,f\left(T;T,p\right)\right\}$$
where,
(6)f(k;T,p)=T−1kpk(1−p)T−kwhere,k∈[1,T]

We prioritize frames with this set of probabilities. Furthermore, we construct a set, $Pr_{n}$, for probability with a binomial distribution, *p*, rather than fixing the probability to a single value. The set $Pr_{n}$ follows:(7)$$\begin{array}{c} Pr_{0}=\left\{\frac{1}{2}\right\}\\ Pr_{n}=Pr_{n-1}\cup \left\{\frac{1}{n+2},\frac{n+1}{n+2}\right\}\\ p∼Pr_{n} \end{array}$$

In other words, the probability, *p*, follows the probability set, $Pr_{n}$, and $F_{p}$ is reconstructed accordingly. The reconstructed final frame selection probability set, $F_{p}^{*}$ is as follows.
(8)$$F_{p}^{*}=\frac{1}{l_{p}}\sum_{p\in {\mathrm{Pr}}_{n}}F_{p}$$
where, $l_{p}$ is length of $Pr_{n}$. However, $F_{p}^{*}$ cannot augment the middle part of the video, which is the keyframe we defined. Therefore, we rearrange $F_{p}^{*}$ around the median to increase the selection probability for the middle part of the video. That is, if the frame order, *k*, is smaller than ⌊T/2⌋, it is sorted in ascending order, and if it is larger, it is sorted in descending order. Therefore, a final set of frame selection probabilities is produced, and based on this, priority probabilities for frame augmentation order are obtained. Figure 5 shows the probability distribution and kurtosis of $F_{p}^{*}$ according to $l_{p}$. As $l_{p}$ increases, kurtosis increases.

#### 3.3.2. Skip Sampling

We chose a sampling method to avoid missing key frames when selecting a frame. Ko et al. proposed a random sampling method that does not lose the keyframe. Therefore, this paper also conducts sampling according to this method. When a fixed size is *N* and the number of frames in the current video is *T*, the average difference *z* of frames follows:(9)$$c=\lfloor \frac{T}{N-1}\rfloor$$

Through *z*, the baseline sequence $S_{n}$ of the video frame follows:(10)$$\begin{array}{c} S_{N}=s+\left(N-1\right)c\;\;where\;\;s=\frac{T-c(N-1)}{2}\\ {\left\{S_{N}\right\}}_{N\in N^{N}} \end{array}$$

A random sequence *R* in the [1, *z*] range is added to the baseline sequence. At this time, the baseline sequence does not exceed the *T*. With this skip sampling method, we do not lose key frames, and the order of frames is maintained.

### 3.4. Sign Language Translation

We used a GRU-based model with a sequence-to-sequence (Seq2Seq) structure with an encoder and decoder in the translation step and improved translation performance by adding attention. This model is widely used in NMT and is suitable for translation tasks with variable-length inputs and outputs. We used Bahdanau attention [32].

Unlike encoding the source sentence in the NMT task, we encode a video’s keypoint vector, $x=(V_{1}^{*},\cdots ,V_{t}^{*},\cdots ,V_{T}^{*})$, and predict the target sentence, $y=(y_{1},\cdots ,y_{i},\cdots ,y_{T_{y}})$. First, the encoder calculates the hidden state and generates a context vector, which follows:(11)$$\begin{array}{c} h_{t}=f\left(v_{t}^{*},h_{t-1}\right)\\ c=q(h_{1},\cdots ,h_{T}) \end{array}$$
where, $h_{t}$ is the hidden state calculated in time step *t* and *c* is the context vector generated through the encoder a nonlinear function. In the decoder, the joint probability is calculated in the process of predicting words, and this follows.
(12)$$p\left(y\right)=\prod_{i=1}^{T_{y}}p\left(y_{i}|\left\{y_{1},y_{2},\ldots ,y_{i-1}\right\},c\right)$$

The conditional probability at each time point used in the above equation is as follows.
(13)$$p\left(y_{i}|y_{1},y_{2},\ldots ,y_{i-1},x\right)=\mathrm{softmax}\left(g\left(s_{i}\right)\right)$$
where, $s_{i}$ is the hidden state calculated from the time *i* of the decoder and follows:(14)$$s_{i}=f\left(y_{i-1},s_{i-1},c\right)$$

The context vector $c_{i}$ is calculated using the weight for $h_{i}$ and follows:(15)$$c_{i}=\sum_{t=1}^{T}=a_{ti}h_{t}$$
where,
(16)$$a_{ti}=\frac{\exp(\mathrm{score}\left(s_{i-1},h_{t}\right))}{\sum_{t=1}^{T}\exp(\mathrm{score}\left(s_{i-1},h_{t}\right)}$$
where, the score is an alignment function that calculates the match score between the hidden state of the encoder and the decoder.

## 4. Experiments

To prove the effectiveness of our proposed method, we conduct several experiments on two datasets: KETI [7], RWTH-PHOENIX-Weather 2014 T [21]. KETI dataset is a korean sign language video consisting of 105 sentences and 419 words and has a full high-definition (HD) video. There were 10 signers, and they filmed the video from two angles. However, the KETI dataset is not provided with a test set, so we randomly split the dataset at a ratio of 8:1:1. The split of training, dev, and test videos are 6043, 800, and 801, respectively. We divided these videos into frames at 30 fps. The KETI dataset did not provide glosses.

The RWTH-PHOENIX-Weather-2014-T dataset is a data set extended from the existing RWTH-PHOENIX-Weather 2014 [23] with the public german sign language. The split of training, dev, and test videos are 7096, 519, and 642, respectively. It has no overlap with the RWTH-PHOENIX-Weather 2014.

The average number of frames of training videos in these two datasets is 153 and 116 frames, respectively. Therefore, the input dataset, *X*, follows $X_{\mathrm{kor}}\in R^{6403\times 153\times 110}$ and $X_{\mathrm{ger}}\in R^{6408\times 116\times 110}$,respectively.

Our experiment was conducted in NVIDIA RTX A6000 and AMD EPYC 7302 16-Core environment for CPU. Our model was constructed using Pytorch [42], Adam Optimizer [43], and Cross-Entropy loss. The learning rate was set to 0.001 and epochs 100, and the dropout was set to 0.5 for preventing overfitting. Finally, the dimension of hidden states was 512.

We tokenize differently according to the characteristics of Korean and German. In Korean, the KoNLPy [44] library’s Mecab part-of-speech (POS) tagger was used, and in German, the tokenization was performed through the nltk library [45]. Finally, our evaluation metric evaluates our model in three ways: BLUE-4 [46], ROUGE-L [47], and METEOR [48] according to the metric of NMT.

Our experiments focused on four aspects. The first aspect involved conducting comparative experiments to evaluate the performance of different normalization methods. The second aspect entailed testing the effectiveness of SASS by experimenting with video sampling and augmentation. The third aspect explored the impact of changes to the length of set $Pr_{n}$ (i.e., $l_{p}$), the representative value *N*, and the sequence of input values. Lastly, we compared our model’s performance to previous studies to confirm its effectiveness.

### 4.1. Effect of Normalization

This experiment proves that our method is robust compared to various normalization methods. The results of this are shown in Table 2. We fixed and experimented with all factors except the normalization method. We compared our method with “Standard Normalization”, “Min-Max Normalization” and “Robust Normalization”. First, standard normalization uses standard deviation, σ, and mean values, μ, of points, *x*, and follows (17). Second, min-max normalization uses the minimum value, $x_{\min}$, and maximum value, $x_{\max}$, of points and follows (18). Finally, robust normalization uses median, $x_{2/4}$, and interquartile range (IQR), $x_{3/4}-x_{1/4}$, and follows (19).
(17)$$X=\frac{x-\mu }{\sigma }$$
(18)$$X=\frac{x-x_{\max}}{x_{\max}-x_{\min}}$$
(19)$$X=\frac{x-x_{2/4}}{x_{3/4}-x_{1/4}}$$

In this case, standard normalization is the method proposed by Ko et al. Moreover, we experimented by adding two normalization methods that did not exist before. First, “all reference” is the normalization method in which (3) is applied to the hand. Second, “center reference” uses only each keypoint vector, *v*, and center vector, *c*, without a separate reference point. This divides the difference between *v* and *c* by distance. Finally, the reference point was compared with the Kim et al. [8] normalization method by fixing it with the right shoulder.

Customized normalization proved to be the most powerful in the experiment. The performance has improved significantly compared to the methods of Kim et al. and Ko et al., which are previous studies. Furthermore, it has proven that it is a new powerful method that can improve performance by performing better in the dataset, KETI, with a relatively large scale frame and in relatively small images, PHOENIX.

All of the methods compared, except our proposed method, have the issue of not properly accounting for the length of human body parts. First, commonly used methods such as Standard, Robust, and Minmax do not account for the fact that people have different body lengths, resulting in poor performance. Secondly, the method used by Kim et al. and the “Center Reference” method, which both use a single reference point, also fail to properly consider the length of each body part. Finally, the “All Reference” method fails to achieve good performance by changing the distribution of hand gestures, which are the most important in sign language. In contrast, customized normalization places a reference point for each body part, allowing the model to understand the length of the human body, and uses min-max normalization for the hand gestures to maintain their distribution, resulting in good performance.

### 4.2. Effect of SASS

This section shows a difference in performance according to frame selection method experiments. Keypoints were normalized with our experimental normalization method, and everything else was fixed except the frame selection method. We demonstrate that our proposed method is excellent with three comparative experiments. We conduct comparison experiments using only sampling, only augmentation, and both sampling and augmentation methods.

To evaluate the impact of frame sampling, we conduct experiments with different numbers of input frames, *N*, for each dataset. The KETI dataset is set to N=50, which aligns with the results of prior research by Ko et al. For the PHOENIX dataset, *N* is set to N=16, the smallest number of frames among the videos in the training dataset, as no similar methods have been reported in previous studies. In this experiment, our goal is to compare the results of frame sampling only, so we set the number of frames to the minimum among the training videos to eliminate any effects of frame augmentation. The skip sampling method showed the best results for the two datasets in the video sampling comparison experiment. Moreover, the stochastic sampling method differs in performance significantly from random sampling following the uniform distribution. While stochastic sampling is more effective than random sampling because it does not miss key frames, skip sampling was more effective because it extracts key frames by taking into account the gap between the frame and the key frame.

Second, we experimented with the performance change when only frames were augmented. We set the maximum frame value of the training video in the dataset to *N* and the *N* of KETI and PHOENIX to 426 and 475, respectively. For the same reason as the experiment that proves the effect of sampling, we set the maximum number of frames of training videos so that frames are not lost (i.e., without the sampling process).

We compared and experimented with a randomly augmented method following an even distribution and a stochastic augmented method following our method. This is shown in Table 3. At KETI, the random augmentation was high performance in every way. However, in PHEONIX, the main evaluation index, BLEU-4, was the best in stochastic augmentation. The overall performance was improved when the frame augmentation was performed compared to when only sampling was performed.

Finally, we experimented with a combination of sampling and augmentation. We experiment with a total of 4 cases. In the previous experiment, since the random sampling method among sampling methods had poor results, we experimented by combining the number of cases except random sampling. This is shown in Table 4. Our method, which combines skip sampling and stochastic augmentation, showed excellent performance in KETI, and BLEU-4 showed the highest performance in PHOENIX.

### 4.3. Additional Experiments

In this section, we experiment with several factors affecting the model. We compare the performance according to the change in the probability set, $Pr_{n}$, and compare it according to *N*. Moreover, we experiment with how to invert the order of input values, a method of enhancing the translation performance proposed in the previous study of Sutskever et al. [30]. For the first time, we experiment with the performance change according to $l_{p}$, the length of probability set $Pr_{n}$. We experiment only with $l_{p}\in \left[3,5,7,9,11,13,15,17\right]$ and evaluate only with BLEU-4. This is shown in Table 5. Our methods outperform other methodologies, including the average BLEU value in the PHOENIX dataset, achieving the best performance on both datasets in the test set with $l_{p}$ of 17.

Second, we experiment with comparison according to *N*. We set and compare *N* according to the training video’s mean and median frame length. The median number of frames of training videos in these two datasets is 148 and 112 frames, respectively. The comparison accordingly is shown in Table 6.

Finally, we experiment with the effect of encoding by inverting the order of input *X*. To this end, we reverse the order of frames and encode each video by changing the order of frames to $\left(V_{T}^{*},V_{T-1}^{*},\cdots ,V_{1}^{*}\right)$. This is shown in Table 7. These results show that setting *N* as the average has the best effect and that entering the frame’s original order has the best effect. It is analyzed that the change in the order of the input sentences is meaningless because our input value is the keypoint value of the frame, not the sentence.

### 4.4. Comparison of Previous Works

In this section, we compared our keypoint and sign language video processing method with a previous study that used the same deep learning model as ours to demonstrate their superiority. In other words, we compared our approach with a study that used an RNN-based Encoder-Decoder model with Attention [13,49]. The KETI dataset was implemented directly and conducted a comparative experiment because we randomly divided it into the training, validation, and test sets. In Ko et al.’s study, additional data were used, and we reproduced and compared the proposed model without adding data, as shown in Table 8.

Moreover, since we present a robust method of experimenting without glosses dataset, the PHOENIX dataset only compares video-to-text translation without glosses. Most recent studies use glosses, so we compare them with Camgoz et al. [4]. This is shown in Table 9.

Our model also demonstrates high performance compared to previous studies in the dataset. Notably, the KETI dataset exhibits the best performance among keypoint based SLT models. Furthermore, the BLEU score was the highest in the PHOENIX dataset. BLEU-4 increased significantly (about 4%p) compared to Sign2Text (Bahdanau Attention), indicating that the ideal performance was improved through our video-processing method even though the same Bahdanau Attention was used.

## 5. Ablation Study

We conduct ablation studies to analyze the effectiveness of our proposed method. Change in performance based on the presence or absence of our normalization and frame selection techniques was experimented with. This is shown in Table 10.

We compare the use of our normalization methods. If customized normalization method was not used, the standard normalization method was used. Moreover, the number of all cases was experimented with by applying different frame selection methods. We only use the skip sampling method for sampling and the stochastic augmentation method for augmentation. Finally, the SASS method, which is our method combining the two methods, is compared when applied.

This experiment proves that we achieve greater performance when using customized normalization and the greatest performance when using SASS.

## 6. Conclusions and Future Work

This paper presents a keypoint-based sign language translation (SLT) model that does not utilize glosses and a novel video processing method that is optimal for this model. We introduce a novel normalization method for keypoint vector processing in sign language videos that are based on human joint positions. Additionally, we propose a method for improving the robustness of our model across multiple datasets by determining the appropriate number of input frames for each dataset rather than using a fixed number of frames for all datasets. Inspired by Camgoz et al. [4], we also demonstrate how the SLT problem can be approached as a machine translation problem.

To demonstrate the effectiveness of our method, we conducted experiments on high-resolution (KETI) and low-resolution (RWTH-PHOENIX-Weather-2014 T) sign language video datasets. The results of these experiments showed that our method is robust across datasets of various sizes. Furthermore, our video-processing method can be applied across various tasks beyond SLT that involve video processing.

In future work, we plan to extend our approach to Sign-to-Gloss-to-Text (S2G2T) translation by leveraging our video-processing method and glosses, as suggested by Camgoz et al. [28] and Yin et al. [25]. We further explore using various neural machine translation models to enhance the performance of our model.

## Figures and Tables

**Figure 1 sensors-23-03231-f001:**
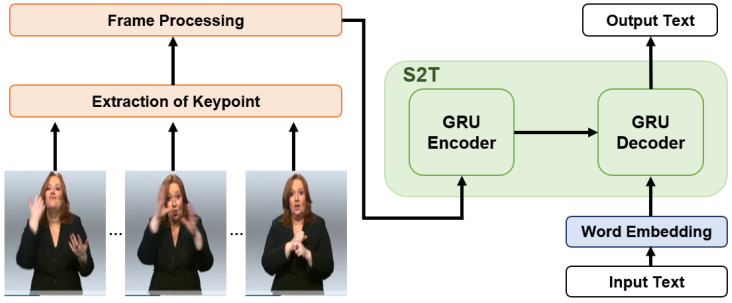
An overview of our approach. S2T means sign-language-to-text model.

**Figure 2 sensors-23-03231-f002:**
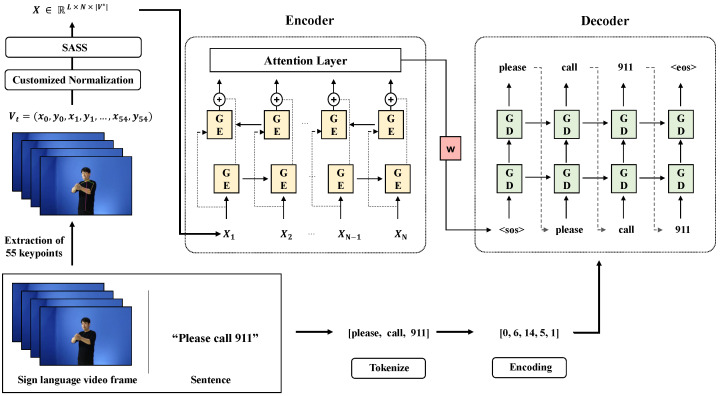
The Full Architecture of our SLT approach. We extract the keypoint vector from each frame and then go through the process of normalizing features. Subsequently, after adjusting the number of frames, encoding is performed. After tokenizing the text, it is put into the decoder through the embedding process. In this case, GE is GRU-Encoder, GD is GRU-Decoder and *X* is the encoding vector that enters the model.

**Figure 3 sensors-23-03231-f003:**
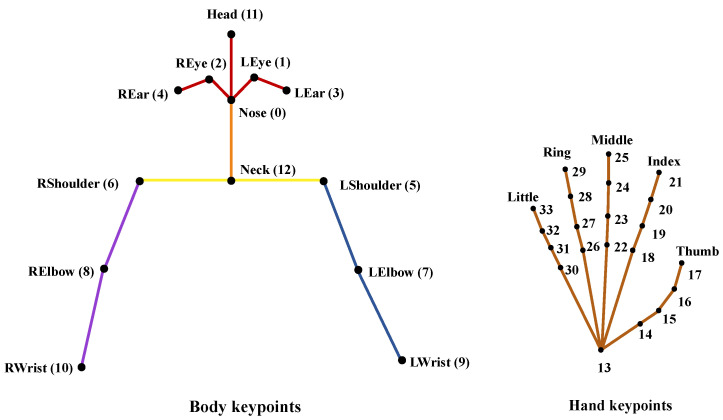
Location and number of the keypoints. We used 55 keypoints and excluded the keypoints of the lower body. Hand keypoints included both left and right hands. Then, we divided each normalization part by color.

**Figure 4 sensors-23-03231-f004:**
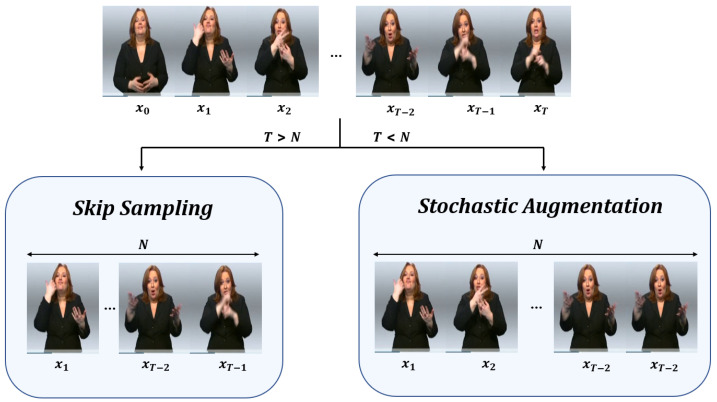
An overview of SASS. This is our frame selection method. If the length *T* of the video frame is less than a fixed scalar *N*, a stochastic augmentation method is used, and if *T* is greater than *N*, skip sampling is used to match *T* with the number of *N*.

**Figure 5 sensors-23-03231-f005:**
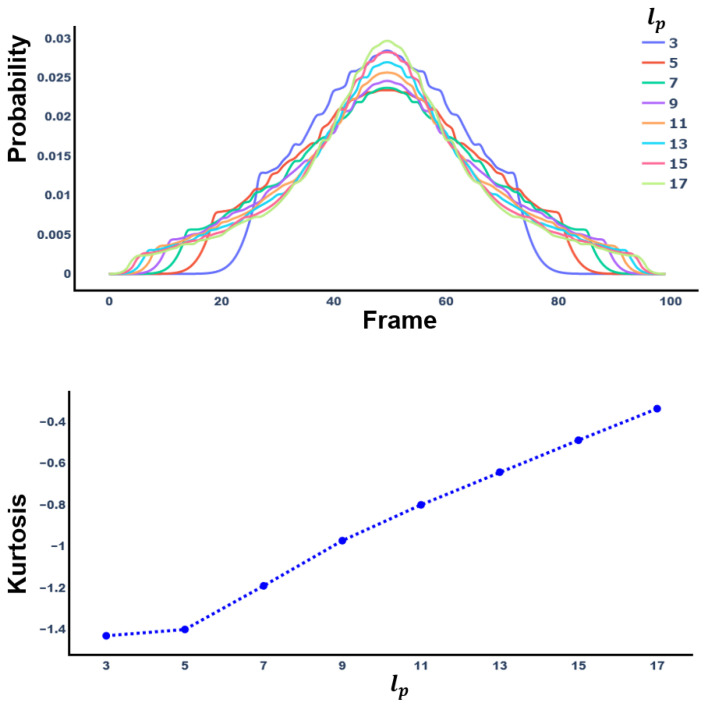
Change of probability distribution and kurtosis following *n*. (**Top**) Probability distribution of probability set, $F_{n}^{*}$, according to the change in $l_{p}$. (**Bottom**) Kurtosis of $F_{n}^{*}$ according to the change in $l_{p}$.

**Table 1 sensors-23-03231-t001:** List of keypoints. We divided them into four parts except for the hands. The number is the index number of the corresponding part, and the reference point is the body part that provides the reference for each part.

Part	Reference Point	Included Number
Face	Nose (0)	0, 1, 2, 3, 4, 11
Upper Body	Neck (12)	5, 6, 12
Left Arm	Left Elbow (7)	7, 9
Right Arm	Right Elbow (8)	8, 10

**Table 2 sensors-23-03231-t002:** Comparison Of Normalization Method.

Method	KETI						RWTH-PHOENIX-Weather 2014 T					
	BLEU-4		ROUGE-L		METEOR		BLEU-4		ROUGE-L		METEOR	
	dev	test	dev	test	dev	test	dev	test	dev	test	dev	test
Standard	78.99	76.44	78.84	76.28	78.84	76.28	12.08	7.36	24.74	23.57	26.93	25.87
Robust	79.17	76.29	79.07	76.23	79.08	76.18	9.19	8.04	24.73	23.57	26.92	25.86
MinMax	79.75	79.01	79.75	78.9	79.75	78.9	9.88	8.05	26.04	24.86	27.47	26.11
Kim et al. [8] method	73.95	72.91	73.8	72.77	73.8	72.77	11.67	10.45	27.2	**26.22**	27.92	27.2
All reference	77.98	76.35	78.09	76.22	78.06	76.22	11.16	9.3	27.08	25.68	28.22	**27.33**
Center Reference	75.65	75.02	75.82	74.81	75.79	74.81	12.1	8.28	**27.66**	25.68	**28.8**	26.79
Ours	**85.3**	**84.39**	**85.48**	**84.85**	**85.58**	**82.31**	**12.81**	**13.31**	24.64	24.72	25.86	25.85

**Table 3 sensors-23-03231-t003:** (TOP) Comparison of SLT Performance by Sampling Method. (Bottom) Comparison of SLT Performance by Augmentation Method.

Method	KETI						RWTH-PHOENIX-Weather 2014 T					
	BLEU-4		ROUGE-L		METEOR		BLEU-4		ROUGE-L		METEOR	
	dev	test	dev	test	dev	test	dev	test	dev	test	dev	test
Random sampling	76.09	75.11	77.17	16.61	77.08	76.45	7.77	5.94	21.74	21.12	23.36	22.63
Skip Sampling	**82.4**	**81.9**	**83.35**	**82.74**	**83.32**	**82.96**	**9.52**	**7.55**	**25.07**	**22.35**	**26.36**	**23.77**
Stochastic Sampling	78.9	78.2	79.92	79.07	79.78	79.08	8.19	6.44	21.45	21.29	22.82	22.86
Random Augmentation	**83.69**	**83.68**	**84.19**	**84.15**	**84.19**	**84.25**	13.71	11.3	**28.11**	27.36	**27.59**	**26.89**
Stochastic Augmentation	82.25	83.33	83.19	80.06	80.74	80.75	**13.74**	**12.65**	28.08	**27.59**	26.36	25.73

**Table 4 sensors-23-03231-t004:** Comparison of combinations of sampling and augmentation.

	KETI						RWTH-PHOENIX-Weather 2014 T					
	BLEU-4		ROUGE-L		METEOR		BLEU-4		ROUGE-L		METEOR	
	dev	test	dev	test	dev	test	dev	test	dev	test	dev	test
Skip + Random	84.88	84.24	85.25	84.75	85.29	84.89	11.64	11.5	27.63	**27.63**	29.69	28.68
Stochastic + Random	83.85	83.92	84.44	84.65	84.4	84.57	10.36	10.16	27.48	26.77	29.47	28.38
Stochastic + Stochastic	84.94	83.5	85.37	83.93	85.45	83.95	9.71	10.83	**27.88**	27.04	**30.01**	**28.7**
SASS (ours)	**85.3**	**84.39**	**85.48**	**84.85**	**85.58**	**85.07**	**12.81**	**13.31**	24.64	24.72	25.86	25.85

**Table 5 sensors-23-03231-t005:** BLEU-4 score by $l_{p}$.

$l_{p}$	KETI		PHOENIX	
	dev	test	dev	test
3	84.38	83.1	11.25	11.17
5	84.49	83.27	12.23	10.48
7	84.4	83.97	12.42	11.74
9	83.41	83.85	**14.84**	12.28
11	84.35	83.45	13.49	12.03
13	84.78	83.31	12.31	12.01
15	84.76	81.68	12.64	11.68
17	**85.3**	**84.39**	12.81	**13.31**
Mean	84.48	83.38	12.75	11.84

**Table 6 sensors-23-03231-t006:** BLEU-4 score by *N*.

*N*	KETI		PHOENIX	
	dev	test	dev	test
Mean	**85.3**	**84.39**	**12.81**	**13.31**
Median	85.22	82.78	12.65	9.27

**Table 7 sensors-23-03231-t007:** BLEU-4 score in order of input values.

Order	KETI		PHOENIX	
	dev	test	dev	test
Reverse	85.1	84.3	12.38	9.13
Original	**85.3**	**84.39**	**12.81**	**13.31**

**Table 8 sensors-23-03231-t008:** Comparison with other models in KETI dataset.

	BLEU-4		ROUGE-L		METEOR	
	dev	test	dev	test	dev	test
Ko et al. [7]	77.96	76.44	77.99	76.56	77.99	76.49
Ours	**85.3**	**84.39**	**85.48**	**84.85**	**85.58**	**85.07**

**Table 9 sensors-23-03231-t009:** Comparison with other models in PHOENIX dataset.

	BLEU-4		ROUGE-L		METEOR	
	dev	test	dev	test	dev	test
Sign2text (Luong) [4]	10.00	9.00	31.8	**31.8**	-	-
Sign2Text (Bahdanau) [4]	9.94	9.58	**32.6**	30.7	-	-
Ours	**12.81**	**13.31**	24.64	24.72	**25.86**	**25.85**

**Table 10 sensors-23-03231-t010:** Ablation Study of Video-Processing Methods. “Normalization” means our customized normalization method. “Sampling” is our skip sampling method and “Augmentation” is our stochastic augmentation method.

			KETI			RWTH-PHOENIX-Weather-2014 T		
**Normalization**	**Sampling**	**Augmentation**	**BLEU-4**	**ROUGE-L**	**METEOR**	**BLEU-4**	**ROUGE-L**	**METEOR**
ine	✓		76.44	76.56	76.49	7.48	21.91	23.75
		✓	77.00	77.62	77.81	11.4	24.62	24.93
	✓	✓	77.78	78.42	78.61	7.36	24.5	**26.21**
✓	✓		81.9	82.74	82.96	7.55	22.35	23.77
✓		✓	80.06	80.74	80.75	12.65	**27.59**	25.73
✓	✓	✓	**84.39**	**84.85**	**85.07**	**13.31**	24.72	25.85

## Data Availability

Our code is available at https://github.com/winston1214/Sign-Language-project.

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
