# Peer review of "Preprocessing for Keypoint-Based Sign Language Translation without Glosses"

_sensors, 2023, doi:10.3390/s23063231_

Round 1
Reviewer 1 Report (Previous Reviewer 1)
The concerns are conveniently addresse.
Author Response
Please see the attachment.

Reviewer 2 Report (New Reviewer)
Preprocessing for Keypoint based Sign Language Translation without Glosses – Review report
After reviewing the article titled " Preprocessing for Keypoint based Sign Language Translation without Glosses", I conclude that the method is worthy of being published. The authors have taken a good effort to structure the manuscript and report the results. However, the manuscript has to be improved.
General
· The article would benefit from language editing as there are quite a lot of pronouns used such as “We”, “ours”, etc.
· Acronyms such as GRU (Line 61), and SASS (Line 124)- are used without providing the full forms.
· Please check the font style used for section 6
· Table 2: check the citation provided with a question mark.
Abstract
This section needs to be improved by highlighting the results of the study and comparison with the existing literature. The Abstract indicated the “qualitative results” however there is no evidence of qualitative results in the manuscript.
Introduction and literature review
The authors have provided a fair introduction and literature review.
Methodology
The authors have done a good effort to present the proposed method. This section is well structured.
However, I highly recommend providing a pseudocode or flowchart to explain sections 3.2 to 3.4. This will enable the readers to comprehend the method more clearly.
Experiments
The experiments are presented well, but the manuscript lacks the analysis and discussion of the results. It is very important to compare and contrast the existing literature with the study results rather than just stating the results obtained for the experiments.
Author Response
Please see the attachment.

This manuscript is a resubmission of an earlier submission. The following is a list of the peer review reports and author responses from that submission.
Round 1
Reviewer 1 Report
General Comments
The paper presents an efficient keypoint-based sign language translation model without gloss. The background, methodology and the results are presented appropriately and comprehensively. The paper is well written. I have a couple of questions and a few suggestions to enhance the clarity for the reader.
Technical Comments
1. The keypoints (including 20 for the second hand) are 53 in total, while 55 are mentioned in the text. Please clarify if 53 in total are used, or otherwise update the Figure 2 showing 55 key points.
2. The keypoint numbers for Right shoulder and Right Elbow seems to be 6 and 8 respectively. Please correct them.
3. Page 11, Line 312: “In this case, we set the number of sampling to N = 50 for the KETI data set and compared it to the Ko et al. method, and in the PHOENIX data set, the minimum frame length N = 16 of the video” How did you select this number i.e. 50 and 16 for KETI and PHOENIX datasets respectively? Was there any experiment? Or relation with the average number of frames for the datasets? Please explain and add in the paper for the reader. Similarly, in Line 319 Page 11, “We set the maximum frame value of the training video in the dataset to N and the N of KETI and PHOENIX to 426 and 475, respectively”. Please explain how did you come up with this number? What is the criterion?
Presentation/Formatting Comments
4. Page 2, Line 72: the last section is supposed to be section 6.
5. In Figure 2, after customized normalization, it should be SASS instead of SPSS.
6. Figure 2, caption “X is the encoded vector that enters the model”. If it is encoded, it should be coming out of the encoder, instead of entering the encoder. Please clarify for the reader.
7. Page 4, line 154, “we used 123 keypoints …..… Therefore, the keypoint vector is composed of…” This is confusing. If you used 55 keypoints, then do not write that we used 123 keypoint at first. Please simplify the sentence to avoid confusion for the reader.
8. Page 10, Line 297, following sentence is duplicated: “This divides the difference between v and c by distance”.
9. I have noticed that some figures/tables (such as Figure 1, Table 1, 2, 3 and 4) are placed earlier than their citation/reference in the text. Please make sure that the figure/table is placed after it is cited in the text.
10. For section heading 3.3, please write the complete name Stochastic Augmentation and Skip Sampling (SASS) instead of SASS only.
Reviewer 2 Report
It was concluded that the original aspects of the study were not sufficient to be published in this journal. Sign language translation is an important field of study, and most of the recent studies in this field have adopted deep learning-based approaches. However, it is a classification process that is currently done in all these studies. There is now a need for studies that propose more versatile and new methods in this regard.
The separation of body and hand key points and subjecting them to deep learning is not a novelty. Anyway, what the CNN architecture basically does is extract these features.
Authors should examine literature reviews and recent studies in more detail. The fact that they presented a superficial point of view in the Related Studies section shows that there is a deficiency in this subject that they need to complete. Since a detailed literature review was not carried out, the comparison with previous studies was also insufficient.